# Red Onion Peel Powder as a Functional Ingredient for Manufacturing Ricotta Cheese

**DOI:** 10.3390/foods13020182

**Published:** 2024-01-05

**Authors:** Florin Daniel Lipșa, Florina Stoica, Roxana Nicoleta Rațu, Ionuț Dumitru Veleșcu, Petru Marian Cârlescu, Iuliana Motrescu, Marius Giorgi Usturoi, Gabriela Râpeanu

**Affiliations:** 1Department of Food Technologies, Faculty of Agriculture, Iasi University of Life Sciences, 3 Mihail Sadoveanu Alley, 700489 Iasi, Romania; flipsa@uaiasi.ro (F.D.L.); roxana.ratu@uaiasi.ro (R.N.R.); ionut.velescu@uaiasi.ro (I.D.V.); pcarlescu@uaiasi.ro (P.M.C.); 2Department of Pedotechnics, Faculty of Agriculture, Iasi University of Life Sciences, 3 Mihail Sadoveanu Alley, 700489 Iasi, Romania; 3Department of Exact Sciences, Faculty of Horticulture, Iasi University of Life Sciences, 3 Mihail Sadoveanu Alley, 700489 Iasi, Romania; imotrescu@uaiasi.ro; 4Research Institute for Agriculture and Environment, Iasi University of Life Sciences, 14 Sadoveanu Alley, 700490 Iasi, Romania; 5Department of Animal Resources and Technology, Faculty of Food and Animal Sciences, Iasi University of Life Sciences, 8 Mihail Sadoveanu Alley, 700489 Iasi, Romania; umg@uaiasi.ro; 6Department of Food Science, Food Engineering, Biotechnology and Aquaculture, Faculty of Food, 13 Science and Engineering, Dunărea de Jos University of Galati, 800201 Galați, Romania

**Keywords:** red onion powder, anthocyanins, antioxidant activity, natural ingredients, value-added products

## Abstract

Onion (*Allium cepa* L.) is a vegetable widely cultivated and consumed due to its rich content in bioactive compounds. Red onion peel (ROP) powder, which is a by-product derived from the onion industry, has been attracting significant interest as a potential functional ingredient for improving the overall quality of foods. The present study explores the potential of ROP powder as a functional ingredient to improve the quality and nutritional value of whey cheese. Despite being frequently viewed as a food processing waste byproduct, ROP is a rich source of bioactive substances, including antioxidants, flavonoids, and dietary fiber, having antioxidant and antibacterial effects. The ROP extract exhibited high amounts of total polyphenols (119.69 ± 2.71 mg GAE/g dw) and antioxidant activity (82.35 ± 1.05%). Different quantities (1 and 3%) of ROP powder were added to cheese formulations, and the subsequent impact on the texture characteristics, sensory attributes, and phytochemical composition of the value-added cheeses was evaluated. The findings show that the addition of ROP powder improved the texture and the color of the cheeses, providing a visually appealing product. Additionally, adding the ROP powder significantly raised the amount of phytochemicals and antioxidant activity (17.08 ± 0.78 µmol TE/g dw for RCROP1, 24.55 ± 0.67 µmol TE/g dw for RCROP2) in the final product’s formulation. Moreover, adding powder to cheese is an effective way to increase the value of onion by-products and produce polyphenol-enriched cheese.

## 1. Introduction

The projected growth in food demand will range from 59% to 98% by 2050 [1]. Farmers across the globe need to increase crop production, either by improving productivity on existing agricultural lands through implementing fertilizer and irrigation practices or by adopting innovative techniques such as precision farming [2]. However, throughout the production process, a multitude of food generates by-products, which subsequently transform into food waste, engendering both environmental and economic ramifications. The by-products possess a high concentration of bioactives, rendering them suitable for potential reuse. Taking into consideration the aforementioned perspective, to adopt the notion of a circular economy, it is plausible to convert waste into novel raw resources [3,4].

Onion (*Allium cepa* L.) is one of the most important vegetable crops worldwide, cultivated since ancient times. It is the second most important horticultural crop after tomato, with onion production increasing by 25% in the last 10 years, with a total global production of around 98 million tons [5]. It has been estimated that only the European Union can produce almost 0.6 million tons of onion waste each year. If not properly disposed of, this waste might damage the environment; due to its strong odor, it cannot be utilized to prepare fodder or as a fertilizer. Despite being a plentiful source of flavonoids and flavanols, onion wastes are still overlooked [6]. 

In addition to being a significant source of pollution and resource exploitation for various industries, this organic waste additionally represents an economic risk. Yet, consumers’ demand for food free of artificial additives [7] and the growing awareness of the circular economy of waste principles implemented in the European Union [8] have prompted academic and industrial research institutes to find alternate uses for onion peels. In powder form, onion peels have been added to several foods of animal and vegetable origins (wheat pasta, gluten-free bread, pizza, yogurt, pork sausages) low in polyphenols and dietary fiber to enhance their functional value, based on its richness in bioactives [9,10]. The bioactive content of onion by-products may be significantly impacted by industrial processing, producing goods with a variety of useful applications. The circular economy is a long-term strategy for lowering environmental effects and offers a long-term solution for preventing, reusing, or recovering natural by-products. This intends to restore by-products to the production process as natural ingredients for new products with considerable health benefits and industry-added value by using sustainable technology to extract nutritive components [11]. Onion peels have been used in the food industry as a natural colorant and a source of dietary fiber. Anthocyanins and quercetin, two flavonoids that have been identified in significant concentrations in onions, have been reported to be especially present in red and yellow onion by-products [12]. In numerous studies, the anticancer, antibacterial, anti-obesity, neuroprotective, cardioprotective, anti-diabetic, and erectile dysfunction activities of onion peel extracts have been highlighted [12,13]. 

The demand for nutraceuticals and functional foods has increased as a result of greater consumer awareness of health and nutrition. The manufacturing of functional foods and nutraceuticals is, therefore, encouraged by researchers and the food industry. Researchers are investigating multiple plant-based ingredients, such as by-products of fruit and vegetable processing (skin, pomace, seeds), as they are rich with phytochemicals, antioxidants, and nutrients and are easily available and reasonably priced [6,14].

In accordance with the tenets of the circular economy, whey is presently employed in the production of several commodities, including whey cheeses, whey powder, edible films, beverages, and feed supplements. Whey has been demonstrated to possess significant value as a product owing to its constituent elements and their functional capabilities. Consequently, various food-related applications, including whey components, particularly proteins, have been devised and implemented. Ricotta cheese is a commonly consumed dairy product that is classified as an unripened acid-heat coagulated-soft whey cheese. It comes from Italy, particularly in the southern region, but it has also gained popularity in the Mediterranean area [15] and has achieved international recognition [16]. Ricotta cheese is a moist and soft dairy delicacy characterized by its moisture content, which usually fluctuates between 65% to 75% [17]. Historically, its foundation lies in the dairy industry’s resourceful use of whey, especially the by-product generated during the crafting of mozzarella [18]. However, it is versatile in its origin, as it can be derived from the whey of various cheeses, including the robust sheep milk cheese or the textured cow milk cheese [19]. While the composition of ricotta can vary depending on regional practices or specific recipes, it is primarily crafted from the whey of sheep or goat or even a blend of the two. However, it is also noteworthy to mention that bovine and buffalo whey contribute distinctive qualities to the final ricotta product [20,21].

From a production viewpoint, ricotta’s distinct texture and taste come from the meticulous heating of whey, followed by the acidification of the heated liquid. This process coagulates the whey proteins, leading to the formation of soft, pillowy curds [17,22]. Once these curds rise to the surface, they are gently scooped and placed into perforated containers, allowing the residual whey (scotta) to drain away, ensuring the cheese’s signature texture [17]. Because agricultural and food by-products are rich in bioactives and have potential nutritional value, they play a significant role in the creation of innovative, sustainable, functional foods as well as the manufacturing of animal feed [23].

In this research, ROP powder was incorporated into whey cheese during its production process, and the impact on its phytochemical, textural, microbiological, microstructure, sensory, and nutritional properties was evaluated. This study highlights the potential of ROP powder as an innovative and sustainable ingredient in the dairy industry, offering the dual benefit of reducing waste and enhancing the nutritional profile of whey cheese.

## 2. Materials and Methods

### 2.1. Reagents and Chemicals

Folin–Ciocalteu reagent, 1% citric acid, 2,2-diphenyl-1-picrylhydrazyl (DPPH), ethanol, 6-hydroxy 2,5,7,8 tetramethylchromane-2-carboxylic acid (Trolox), potassium chloride solution, sodium hydroxide, sodium nitrite, Gallic acid, sodium acetate solution, aluminum chloride, sodium carbonate, were obtained from Sigma Aldrich Steinheim (Darmstadt, Germany). All other reagents used in the experiments were of analytical grade.

### 2.2. ROP Powder Preparation

Red onions (*Rosie de Aries*) were bought from a supermarket (Iasi, Romania) in June 2022. The outer layers of the red onions were removed, washed with distilled water, and dried for two hours at 40 °C, up to a moisture content of 10.0%, in a standard oven (Stericell 111, MMM Medcenter, München, Germany). The ROP was pulverized (mean particle diameter of 1 mm), kept at room temperature in an airtight glass jar, and utilized for further extraction. The ROP powder underwent decontamination through sterilization using a UV lamp.

### 2.3. Extraction of Bioactive Compounds from ROP Powder

For the extraction of the bioactive compounds from red onion peel powder, the method using ultrasound described by Albishi et al. [24] was used, with slight modifications. An amount of 1 g of ROP powder was solubilized with 14 mL of 70% ethanol acidified with 1% citric acid solution in a ratio of 1 to 13 of acid/solvent. The samples were sonicated using a sonication bath (Elmasonic S 180 H, Elma, Singen, Germany) for 40 min at 40 kHz, at 25 °C, and then centrifuged for 10 min at 6500 rpm and 4 °C, the supernatant being phytochemically characterized.

### 2.4. Extract Characterization

For the red onion peel extract, the anthocyanin contents, flavonoids, polyphenols contents, and DPPH radical scavenging activity were determined.

#### 2.4.1. Determination of Total Anthocyanin Content (TA)

The TA content of ROP extract was determined spectrophotometrically with an Analytik Jena (Specord 210 Plus, Jena, Germany) UV-Vis spectrophotometer, using diluted samples (1:10), according to the modified protocol used by Giusti and Worsltad [10]. 

A mixture of 200 μL of the sample and 800 μL pH = 1.0/pH = 4.5 buffer solution was added in the spectrophotometer cuvette, and, after a 15 min rest in the dark, during which the solutions reacted, the absorbance was read at 520 and 700 nm. The AT content expressed in mg cyanidin 3-glucoside equivalents (C3G)/g dry substance (dw) was calculated as follows (1): (1)$$TA\;(mg\;C3G/g\;dw)=\frac{A\times Mw\times DF}{\varepsilon \times l\times m}$$
where A = (A520 nm–A700 nm) pH 1.0–(A520 nm–A700 nm) pH 4.5; Mw is 449.2 g/mol, the molecular weight of cyanidin-3-glucoside; Df is dilution factor; l = 1 cm is cuvette path length; m is the amount of sample; ε = 26,900 L/mol/cm is molar extinction coefficient for cyanidin-3-glucoside 

#### 2.4.2. Determination of Total Polyphenol Content (TP)

To determine the TP content, the Folin-Ciocâlteu colorimetric method was used. For this method, described by Horincar et al. [25], 0.20 mL of extract was combined with 15.8 mL of distilled water and 1 mL of Folin-Ciocâlteu reagent. After 10 min, 3 mL of 20% sodium carbonate solution was added. The mixture was incubated for 60 min at room temperature, and then the absorbance at 765 nm was read. The content of polyphenolic compounds was expressed as mg gallic acid equivalents (GAE)/g dw by means of an equation from the standard Gallic acid calibration curve (y = 1.6991x − 0.0256 with R^2^ = 0.9837).

#### 2.4.3. Determination of Total Flavonoid Content (TF)

The method used by Albishi et al. [24] with some modifications was used to determine the TF content of the supernatant from ROP. A volume of 0.500 mL of the diluted extract (1:10) was mixed with 0.150 mL of 5% sodium nitrite and 2 mL of distilled water. The mixture was allowed to react for 5 min, after which 0.150 mL of 10% aluminum chloride was added. The mixture was left to react for 6 min. Then, 1 mL of 1 M sodium hydroxide was added. The absorbance of the resulting mixture was immediately measured at a wavelength of 510 nm, the TF content being expressed in mg quercetin equivalents (EQ)/g dw, using the equation from the standard quercetin calibration curve (y = 1.46x − 0.008 with R^2^ = 0.9978).

#### 2.4.4. Determination of the Antioxidant Activity

To determine the antioxidant activity of the extract, the method using DPPH (2,2-diphenyl-1-picrylhydrazyl) described by Horincar et al. [25] was used, with minor modifications. A volume of 3.9 mL of DPPH solution reacted with 100 µL of diluted sample for 30 min at room temperature in the dark. For the blank, 3.9 mL of DPPH was mixed with 100 μL of methanol. The absorbance of the solution was read at 515 nm. The antioxidant activity of the extract was expressed in µmol Trolox equivalent (TE)/g dw by reference to the calibration standard curve with Trolox with the equation y = 0.45x + 0.0075 and R^2^ = 0.993. The radical scavenging activity was expressed also as the percentage of inhibition based on Equation (2): (2)$$\mathrm{DPPH}\;\mathrm{scavenging}\;\mathrm{activity}\;(\%)=\frac{Abs\;Control-Abs\;Sample}{Abs\;Control}\;\times \;100$$
where Abs Control is the absorbance value of the DPPH solution only, and Abs Sample is the absorbance value of the DPPH solution mixed with ROP extract. 

### 2.5. Preparation and Characterization of Supplemented Ricotta Cheese

In order to obtain value-added ricotta cheese, the ROP powder was used in two different ratios (1% and 3%) to test its functionality. The control sample (RCC) consisted of whey cheese without added powder. The final choice of the respective 1 and 3% concentrations of ROP powder used for the fortification of the studied dairy product had taken into account the positive sensory evaluation of the panelists and the best quality in terms of appearance, texture, taste, and color.

The ricotta-type cheese was made using the whey derived from the manufacture of the cheese. The production technology employed the process of agglomeration and precipitation of the whey protein, facilitated by acidification. This was achieved by maintaining pH levels between 6 and 5.8 and heating the mixture to temperatures ranging from 80 to 90 °C. The final product was obtained within 10 to 30 min, depending on the specific heating method used. Subsequently, the coagulated whey was gathered in molds to separate the exhausted whey (scotta) and then underwent a cooling process. Once the product had been chilled, three batches were created by incorporating onion peel powder using a Voltz Planetary Mixer with a power output of 2100 W. Figure 1 provides a description of all the stages involved in the technical flow.

### 2.6. Characterization of Phytochemicals and Antioxidant Activity of the Supplemented Ricotta Cheese Samples 

The total anthocyanins, flavonoids, phenolic content, and antioxidant activity of ricotta cheese enhanced with ROP powder were assessed using the techniques previously described in Section 2.4. To study the storage stability, the samples were stored in light-resistant glass containers at 4 °C for 21 days, and the changes in the phytochemical content were recorded.

### 2.7. Color Evaluation of Supplemented Ricotta Cheese Samples

Using a MINOLTA Chroma Meter model CR-410 (Konica Minolta, Osaka, Japan) with a CIE Lab scale, the color characteristics of the samples were assessed. The results of the color measurements were expressed as L*, lightness (black: L* = 0 and white: L* = 100), a*, ranging from red to green, and b*, ranging from yellow to blue. Following equipment calibration against a white plate, the CIELAB color parameters were collected in triplicate. The hue angle (Hue angle = 180 + arctan(b*/a*) was also determined for quadrant II (−a*, +b*)), which describes the visual color appearance, the Chroma ($\sqrt{{{(a}^{*})}^{2}+{{(b}^{*})}^{2}}$), which describes color intensity, and ΔE ($\sqrt{{L^{*}}^{2}+{a^{*}}^{2}+{b^{*}}^{2}})$ the total color difference [26]. 

### 2.8. Textural Parameters of Supplemented Ricotta Cheese Samples

To check the textures of the samples, a Mark 10 ESM 300 texturometer (New York, NY, USA) with a 7i-50 series digital dynamometer (with a measurement resolution of 0.05N) was used. The cylindrical probe used for the compression tests was model TA4, with a diameter of 38.1 mm and a height of 20 mm. The warp samples subjected to texture analysis had a cylindrical shape with a diameter of 30 mm and a height of 40 mm. The texture tests were performed by applying double compressive stress to the warp samples with the cylindrical probe, obtaining a characteristic force-versus-time diagram. Three repetitions were performed for all texture trials.

### 2.9. Sensory Evaluation of Supplemented Ricotta Cheese Samples 

Ten untrained tasters (aged 24–65, 60% women and 40% men) assessed the sensory quality of fortified cheese samples. The panelists were informed of the overall purpose of the study as well as the necessary procedures for handling personal data. The panelists were asked to evaluate 12 descriptors, including appearance, section appearance, odor, aroma, hardness, adhesiveness, color, taste, chewability, aftertaste, and the overall assessment. The analysis was carried out in accordance with ISO 13299 (2016) [27] requirements. According to Faccia et al. [28], the assessors assigned a grade to each attribute based on a seven-point hedonic scale (1 = extremely low; 7 = extremely high). The Ethics Committee of the Faculty of Agriculture, University of Life Sciences in Iasi, Romania, confirms that approval has been issued for the sensory examination of ricotta cheese with and without ROP powder. Group members were informed about the objectives of the study and the handling of personal data. Informed consent forms were provided that clearly outlined the voluntary nature of participation, the right to withdraw at any time, and their confidentiality.

### 2.10. Microbiological Analyses

All analytical procedures regarding the enumeration of the microbial load of ricotta cheese were conducted in a sterile environment with 3 replicates for each laboratory sample. For the microbiological analyses, 10 grams of cheese were homogenized with 90 milliliters of buffered peptone water (Bio-Rad, Marnes-la-Coquette, France) in a laboratory blender (Seward, West Sussex, UK) for 5 min at 250 rpm. The spread and pour plate techniques were used to isolate the bacteria, yeast, and molds from the serial dilutions, which were made by mixing 1 mL of the previous dilution with 9 mL of buffered peptone water in test tubes [29].

The microbiological media used for plating after inoculation with 1 mL of sample were the non-selective Plate Count Agar (PCA) added with 1 g/L skimmed milk powder and Potato Dextrose Agar (PDA; Scharlau, Barcelona, Spain), plus the selective chromogenic agar Rapid *E. coli* 2 (RE) and Rapid Staph (RS; Bio-Rad, Marnes-la-Coquette, France). Total aerobic bacteria were counted after incubating inoculated PCA plates for 72 h at 30 °C, in accordance with ISO 7218 [30]. Yeast and molds were enumerated after cultivation on PDA plates at 28 °C for 5 days. Colonies of *Escherichia coli* and other coliforms were determined according to the ISO 16140 [31] on RE medium after incubation at 37 °C for 24 h. All samples were also cultured on RS medium, which guarantee the detection and enumeration of coagulase-positive staphylococci (*Staphylococcus aureus*) in 24 h at 37 °C. 

After incubation, microorganisms from the cheese samples were counted using the automatic colony counter Scan 1200 (Interscience, Saint-Nom-la-Bretèche, France), and results were expressed as logarithmic colony forming units per gram (log CFU g^−1^).

### 2.11. Scanning Electron Microscopy Analysis

The samples were subjected to morphological characterizations using a scanning electron microscope (SEM) (Quanta 450, FEI, Thermo Fisher Scientific, Hillsboro, OR, USA) along with an energy dispersive X-ray detector (EDS) (EDAX, AMETEK Inc., Berwyn, PA, USA). The EDS spectra analysis was conducted using the TEAM version V4.1 system developed by EDAX Inc. Before the study, a calibration was performed using a standard AlCu sample consisting of a copper foil placed on an aluminum grid. The samples were analyzed in low vacuum conditions, with a pressure of about 6.1 e^−4^ Pa, while the acceleration voltage of the electrons was 15 kV and viewed at a magnification of 500× (20 μm).

### 2.12. Statistical Analysis

Data reported in this study are mean values ± standard deviation of the mean and represent the means of triplicate analyses. Statistical data analysis was performed using the Data Analysis Toolkit in Microsoft Excel software (Microsoft Office LTSC Professional Plus 2021-Version 2108, Build 14332.20615). Significant differences between samples were quantified using one-way ANOVA after checking the conditions of normality and equality of variances. Tukey’s post hoc analysis was performed considering a 5% significance level (*p* < 0.05) (Minitab Inc., State College, PA, USA).

## 3. Results and Discussion

### 3.1. Phytochemical Characterisation of ROP Powder

Before cheese production, ROP was characterized for its phytochemical profile, and the extract exhibit a remarkable content of bioactive compounds with good antioxidant activity. The ultrasound-assisted extraction method was performed to extract bioactive compounds from ROP in 70% ethanol acidified with 1% citric acid. The phytochemicals and color characteristics of ROP powder are presented in Table 1.

ROP powder displayed excellent quantities of antocyanins 2.47 ± 0.09 mgC3G/g dw along with a DPPH scavenging activity 42.29 ± 1.86 µmol TE/g dw and a total phenols and flavonoids content of 119.69 ± 2.71 mg GAE/g dw and 164.66 ± 2.93 mg QE/g dw, respectively. The changes in color L*, a*, and b* of ROP powder were 33.08 for L* value, a* value 21.93, and b* value 4.51. Chroma, hue angle, and total color change (ΔE) values were calculated from color parameters as 22.39, 0.21, and 39.94. The powder’s color indices’ results showed that it was situated in the first quadrant (+a*, +b*). In their study, Katsampa et al. [6] discovered that the ROP extract obtained by the ultrasound-assisted extraction method, utilizing a 90% (*w*/*v*) aqueous glycerol as the solvent, contained lower antocyanins 1.87 ± 0.39 mg C3G/g dw and polyphenols 61.47 ± 14.19 mg GAE/g dw under sonication (140 W, 37 kHz), at 45 °C, for 60 min. Viera et al. [32] acquired a higher TP of 270.46 mg GAE/g dw and antocyanins of 3.24 mgC3G/g dw obtained through ultrasound in pulsed mode at 130 W and 60% ethanol for 20 min at 25 °C. Depending on the variety, the agronomic circumstances of the area where they were grown, the extraction methods employed (e.g., type of solvent, temperature, pH, and light intensity), and the measurement methods applied, the composition of onion peel varies. Up to this point, it has been determined that ROP extracts are a good source of phytochemicals with antioxidant activity and may be used as ingredients in foods, aiding in reducing agro-industrial residues. 

According to a preliminary High-Performance Liquid Chromatography (HPLC) analysis, which was also reported in our earlier study [33], cyanidin 3-O-laminaribioside and cyanidin 3-O-(6′′-malonyl-laminaribioside) were the two main anthocyanins identified in the ROP extract. Furthermore, other phenolic compounds were detected, with quercetin and catechin being the predominant polyphenols that were tentatively identified. Our findings concur with other research that Sharif et al. [34] and Celano et al. [35] presented. 

Our results highlighted the potential of ROP powder from the *Rosie de Aries* cultivar to be used as a natural additive for dairy products. ROP is a rich source of polyphenols, which are the most abundant antioxidants found in the human diet and are associated with many possible health advantages [36]. For this purpose, a ROP by-product was applied at 1% and 3% during the fresh bovine cheese production. 

### 3.2. Characterization of Bioactive Potential of Supplemented Cheeses and Storage Stability of the Samples

Food manufacturers have recently taken into consideration using naturally occurring antioxidants that are derived from plant by-products to meet consumer demand for more functional and healthful diets [37]. In particular, concerning dairy products, there has been a growth in the enrichment of cheeses with various unconventional ingredients that include physiologically active substances and a balanced nutritional profile [38]. Phenols are the most remarkable antioxidants found in fruits and vegetables [39]. Because of this, there is increasing interest in the bioactive phenolic compounds found in the by-products of red onions.

In order to highlight the added value of cheese, different percentages of ROP powder of 1% (ricotta cheese with 1% ROP powder -RCROP1%) and 3% (ricotta cheese with 3% ROP powder RCROP3%) were added, and the phytochemical characterization and the stability of the bioactive compounds was monitored during 21 days of storage. 

The phytochemical profile and antioxidant activity of control and experimental cheeses are reported in Table 2. According to the Tukey test, statistically significant differences (*p* < 0.05) were found among enriched cheeses. The findings showed that adding ROP powder to the ricotta cheeses significantly increased the TA, TP, and TF contents of the samples. The concentrations of phytochemicals and antioxidant activity significantly increased (*p* < 0.05) with the powder content from 1% to 3%. Additionally, a positive and direct correlation was found between the powder concentration and the antioxidant activity, meaning that increasing the concentration of ROP from 1% to 3% also significantly increased antioxidant activity (*p* < 0.05). The presence of certain phenolic compounds in the ROP powder is most likely what caused the increase in antioxidant activity. Their retention in the cheese curd affects the compounds’ ability to be recovered in cheese.

Over time, the TA, TP, and TF contents of the samples decreased (*p* < 0.05). Nevertheless, the enriched cheese’s phytochemicals analyzed remained higher than the control. The TP of the cheeses was in the range of 4.25–6.05 mg GAE/g dw of fortified cheese on the first day and reached 3.06–5.38 mg GAE/g dw of fortified cheese on the last day of storage. Also, after 21 days, all samples showed a decline in DPPH scavenging activity, while RCROP1% and RCROP3%’s activity remained higher than control. A cheese fortified with chamomile extract showed a comparable decline in shelf life, according to Caleja et al. [40]. The authors explained that the extract degradation during storage leads to loss of bioactivity. 

Increasing the supplementation of ROP powder from 1 to 3% has demonstrated excellent improvement in TA, TP, and TF contents in supplemented cheese when compared to control samples. The addition of ROP powder resulted in enhanced DPPH scavenging activity in cheeses, with a notable increase seen, particularly in the case of RCROP3%. Han et al. [41] and Giroux et al. [42] have both reported similar results in their studies on cheeses that were enriched with natural plant extracts and green tea extracts, respectively. Hamdy and Hafaz [43] reported that ricotta cheeses enhanced with thyme, basil, and rosemary had higher phenol content and antioxidant activity than the control cheese. Additionally, throughout the storage period, all samples showed a decline in phenol content and antioxidant activity.

The findings depicted in Table 2 confirm the enhanced quality of cheeses when supplemented with ROP powder, as seen by the elevated levels of anthocyanins and polyphenols. These compounds contribute to a cheese product with enhanced antioxidant activity. The findings of this study indicate that the utilization of ROP powder has the potential to serve as an alternative to synthetic colorants and antioxidants.

### 3.3. Color Evaluation of Supplemented Cheese Samples

ROP powder presents a promising possibility to enhance the quality of whey cheese by improving its texture, flavor, and nutritional content. The assessment of color is an important factor in determining the quality of food products since it is closely associated with desirability and overall food quality [44]. 

The results of the color attributes (L*, a*, b*) after obtaining the fortified cheese samples and after 21 days of storage at 4 °C are revealed in Table 3. 

The analysis of the color parameters revealed that the addition of ROP had a significant impact on the color of the cheeses. Specifically, the experimental cheeses had the highest redness (a*) values and the lowest lightness (L*) and yellowness (b*).

The total color change attribute, or ΔE, had a range of 52.56 to 51.93 for the RCROP3% sample during storage for 21 days. The addition of ROP powder led to a significant (*p* < 0.05) increase in ΔE during storage. The color’s Chroma, which expresses its intensity and saturation, was at its highest in RCROP3% cheese. Since hue angles were less than 10°, the hue angle value was proportionate to the received color and demonstrated the redness of both enriched samples. Generally, a red hue is represented by an angle of 0° or 360°, while a yellow, green, or blue hue is represented by an angle of 90°, 180°, or 270° [45]. 

The addition of ROP powder had a substantial impact on the lightness of cheeses, and a statistically significant increase was observed in terms of redness. Similarly, a decline in L* values was noted by several researchers following the inclusion of plant extracts in their studies [40,42]. The red value (a*) of cheeses supplemented with ROP powder gradually increased with the addition of ROP powder. When compared to control samples, the other color characteristics for white L* and yellow b* of the enriched cheese steadily declined as the amount of ROP powder increased. 

As anticipated, the incorporation of ROP resulted in substantial modifications to both the external and internal color indexes. The experimental cheeses displayed a noticeable burgundy hue, as evidenced by a significant redness increase, corresponding to a decrease in lightness and yellowness. 

### 3.4. Textural Properties of Supplemented Cheese Samples

The texture parameters of supplemented cheese, such as firmness, adhesion, cohesiveness, gumminess, springiness, chewiness, and resilience, are presented in Table 4. 

The minimum value recorded for firmness was in the control sample on the first day of determinations and after 21 days of storage. Therefore, it is found that the incorporation of ROP contributes to the observed increase in cheese paste firmness as assessed by its compressive strength. This result is probably attributed to the influence of ROP on increasing the dry matter or reducing the fat content of the experimental cheeses. It is found that the value of the firmness of the cheese paste decreases during storage as a result of its moisture loss and the biochemical changes that occur over time. Regarding the gumminess, an increase in its values was observed by adding the powder compared to the control samples, but during storage, the value decreased due to the decrease in fat content, suggesting that the action of proteases and/or irreversible denaturation more frequently weakens the protein bonds. A reduction in resilience during storage, or the ability of cheese samples to return to their original shape after the first compression, was also observed. The statistical analysis did not reveal significant variations (*p* > 0.05) in the control cheeses both immediately after production and throughout the storage period. Similarly, no notable distinctions were observed between cheeses that included ROP powder.

Ensuring a consistent texture profile throughout the shelf life is crucial to both marketing success and sensory acceptance. The protein matrix of the cheese undergoes a transformation from a spongy texture to a denser structure. Thus, as a result of this fact, an increase in the values of the adhesion, cohesiveness, springiness, and chewiness texture parameters is found during ripening for all the analyzed samples. 

The firmness of the experimental cheeses exhibited a statistically significant increase (*p* < 0.05) when compared to the control sample. The phenomenon described can be further explained through the process of polyphenol binding to casein, resulting in the fortification of the curd network and subsequent enhancement of the cheese firmness. Giroux et al. [39] showed comparable findings in their study on cheeses that were supplemented with green tea extract. According to them, the impact of polyphenols on cheese texture can be attributed to their ability to decrease moisture content and influence the structural arrangement of the paracasein matrix. Similar trends were noted for the springiness and adhesiveness properties, as they exhibited an increase in the fortified samples. The fortification did not have an impact on cohesion. Giroux et al. [42] reported varying results, specifically a decrease in cohesiveness and springiness.

### 3.5. Microbial Quality in Supplemented Cheese Samples

Table 5 displays a summary of the microbiological analysis for the prevailed samples from ricotta cheese without red onion peel powder (ROP, control sample) and value-added ricotta cheese with ROP used in two different ratios (1% and 3%), indicating that the product was safe for consumption. 

All samples exhibited pathogen absence for *Escherichia coli* and coagulase-positive staphylococci (*Staphylococcus aureus*). Coliforms were detected in all cheese samples, and their occurrence was not statistically significant, comparing the control sample to 3% ROP value-added cheese, with mean concentration values ranging from 2.47 to 2.65 log CFU/g.

The pH, salt content, type, and quantities of added fortifying compounds have an impact on the presence and quantity of microorganisms in foods, such as fortified cheeses. Sharma K. et al. [11] reported that red onion, one of the best natural sources of quercetin with activity against free radicals, also possesses anti-fungal and antibacterial properties conferred by allicin (thiosulfinates), which is a sulfur-containing volatile compound and exists in onions.

The highest viable microbial counts (i.e., total aerobic bacteria, yeast, and molds as well as total coliforms) were reported in the case of 3% ROP fortified cheese compared with the control cheese sample, but the recorded data are within the permitted limits allowed by European Regulation 2073/2005.

### 3.6. Microstructure Analysis

Figure 2 displays the scanning electron microscopy (SEM) pictures of the ricotta cheeses. The control cheese, which was not supplemented with ROP powder, had a dense protein structure characterized by a granular and irregular surface. This observation suggests a decrease in the porous consistency of the cheese porous consistency and a corresponding increase in its density.

The addition of ROP powder into cheeses was hypothesized to have disrupted the protein matrix and induced molecular reorganization of proteins, leading to the formation of stronger bonds [41]. This was supported by the observed increase in cheese hardness, as indicated in Table 4. The cheese samples exhibited variations in microstructure, while the matrix of the RCROP1% and RCROP3% cheeses demonstrated a heterogeneous composition characterized by a fine-linked network. The presence of casein micelles was more pronounced in the aforementioned samples in comparison to the control cheese. The results of RCROP3% indicated a structure that was both more compact and denser.

In general, the cheeses enhanced with ROP powder demonstrated a greater level of cross-linking and stronger linkages. The enhanced hardness and deformability observed in the texture profile of these cheese samples can be attributed to the presence of a more condensed matrix. Additionally, the SEM analysis demonstrated that cheese samples containing higher concentrations of ROP powder exhibited greater protein aggregates inside their matrices in comparison to the control group. Consequently, the casein clusters exhibited greater thickness and compactness inside the networks of RCROP3% cheeses compared to other samples, an effect that can be ascribed to the elevated ROP concentration. The incorporation of ROP powder into cheese led to an enhanced consistency and uniformity in both texture and structure, attributed to the presence of dietary fiber within ROP. The results correspond with the findings of Mourtzinos et al. [46] and El Hatmi et al. [47], who observed that incorporating onion wastes and *Allium roseum* powder into yogurt and soft cheese had a beneficial effect on the texture of the products.

### 3.7. Sensorial Analysis of Supplemented Cheeses

Sensory evaluation of the formulated cheeses was realized using a seven-point hedonic scale. The average scores obtained from the sensory evaluation are displayed in Figure 3. 

The addition of 3% ROP did not significantly change the taste of the cheese (*p* > 0.05). Initially, the RCROP1% and RCROP3% samples exhibited a drier granulated appearance with a light and fresh taste. In the same way, there was no discernible difference in the odor descriptor scores across the cheeses. In general, the aroma scores were still above the acceptable level, in which a score of >5 was obtained. Although Table 5 previously demonstrated that the fortified samples were firmer than the control, the addition of ROP powder did not significantly influence hardness scores. This suggested that the panelists could tolerate the fortified samples’ increased hardness. Perceived variations between products do not always have an impact on their acceptability, according to Costell et al. [48]. 

The diagram depiction unequivocally demonstrates that adding 1% to 3% ROP powder resulted in a type of cheese with distinct sensory characteristics. When comparing different additions with 1 and 3% ROP powder, the cheese with 1% scored roughly closer to the control samples in all sensory qualities. Cheese containing 1% and 3% ROP powder demonstrated higher scores for sensory parameters compared to control. 

The color was determined as satisfactory as the control for the samples RCROP1 and RCROP3%. In the present study, it was observed that the incorporation of ROP powder into cheese samples resulted in a noticeable enhancement of the burgundy hue (Figure 4). When the concentration of ROP powder was 3%, the panelists observed a highly pronounced red color, which resulted in the highest acceptance score. Finally, the experimental cheeses demonstrated the greatest overall assessment score, sometimes referred to as the degree of overall satisfaction. The study revealed that the panelists appreciated the red hue of ricotta cheese supplemented with onion powder. Furthermore, it was observed that the acceptability, flavor, color, and texture of the cheese increased throughout storage. Overall, all the samples of cheese enriched with red onion powder exhibited favorable sensory evaluations. However, the sample containing 3% ROP powder demonstrated the highest level of performance after the chosen storage duration.

The panelists assessed the sensory characteristics of the experimental and control cheeses and pointed out several variations between them. Nonetheless, it is commonly recognized that the taste characteristics of cheeses are significantly impacted by the inclusion of ROP or other by-products [38,49]. Experimental cheeses showed a higher complexity of odor and aroma intensity. In terms of the panelists’ overall evaluation, ROP-enriched cheese received the highest rating, indicating that the incorporation of vegetable by-products into dairy products positively impacted their sensory qualities and led to the development of novel dairy products [50]. In comparison to other cheeses and control cheese, the sample containing 3% ROP powder achieved the greatest scores for its organoleptic characteristics. These findings demonstrated that, at the end of cold storage, all samples treated with ROP powder outperformed the control sample. According to research by Hamdy and Hafaz [43], adding thyme, basil, and rosemary to ricotta cheese improved its sensory qualities. Throughout the cheeses’ 21-day storage, there was a small decrease in the cheeses’ sensory acceptance. El-Den (2020) [51] discovered that adding curcumin to ricotta cheese preserved the cheese’s sensory appeal while they were being stored.

This functional component may be used in whey cheese formulations to meet customer demand for dairy products that are more appealing and healthful while also offering an environmentally beneficial way to reuse agricultural by-products.

## 4. Conclusions

The results highlighted that the ROP extract is an important source of bioactive compounds with high antioxidant activity. This investigation showed that adding 1% of ROP powder to ricotta cheese did not interfere with the commercial starter LAB’s fermentation process. Furthermore, this enrichment made it possible to produce a bovine whey cheese with particular functional, physicochemical, and sensory qualities. Higher protein, higher polyphenols, and lower fat content were the characteristics of ROP-enriched cheese. The results indicated that the RCROP2 cheese stood out in terms of satisfactory texture, sensory, and phytochemical properties with significantly high amounts of bioactive compounds, suggesting the functional potential of ROP. Therefore, this dairy product seems suitable to support the human body’s ability to maintain the proper balance of important nutrients, and it may also be a promising resource in the market for innovative foods. The addition of ROP powder to whey cheese creates new possibilities for product development and market differentiation as consumers seek healthier and more useful foods. To maximize the advantages of this functional component, more investigations are necessary to determine the optimum inclusion rates and processing settings for ricotta cheese manufacturing. 

## Figures and Tables

**Figure 1 foods-13-00182-f001:**
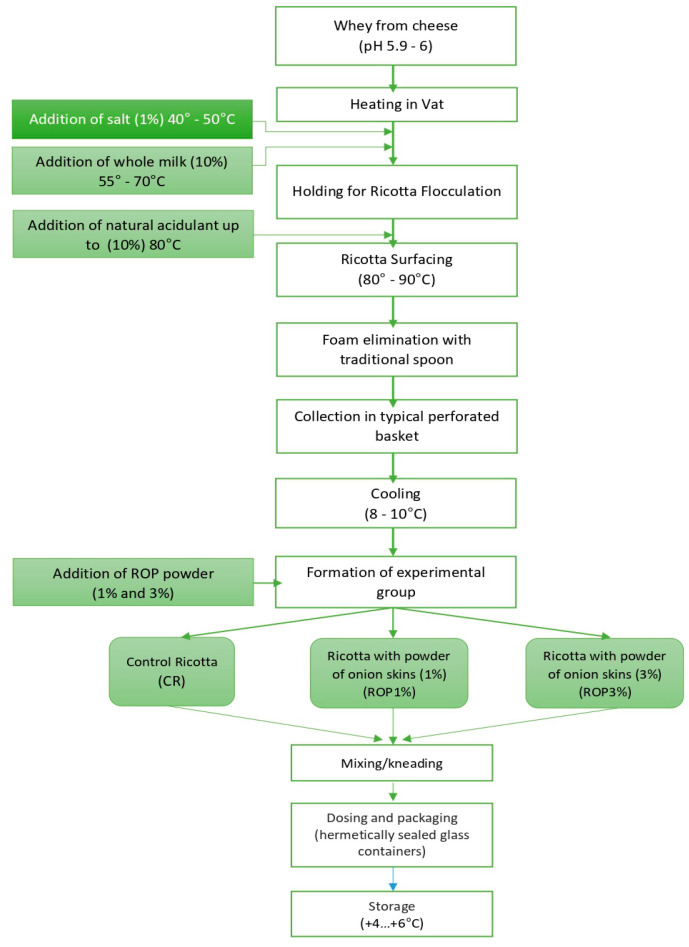
Flow chart of ricotta cheese production.

**Figure 2 foods-13-00182-f002:**
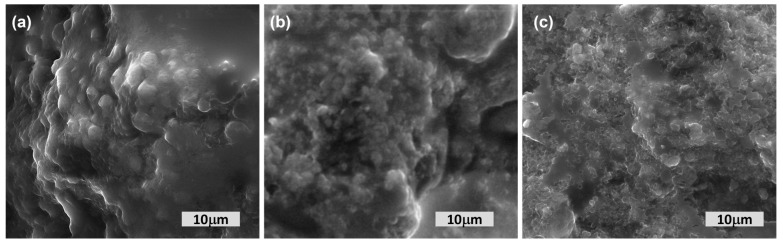
SEM micrographs of whey cheese enriched with ROP powder (**a**) RCC cheese without powder addition; (**b**) RCROP1% and (**c**) RCROP3% cheese with 1 and 3% powder of ROP.

**Figure 3 foods-13-00182-f003:**
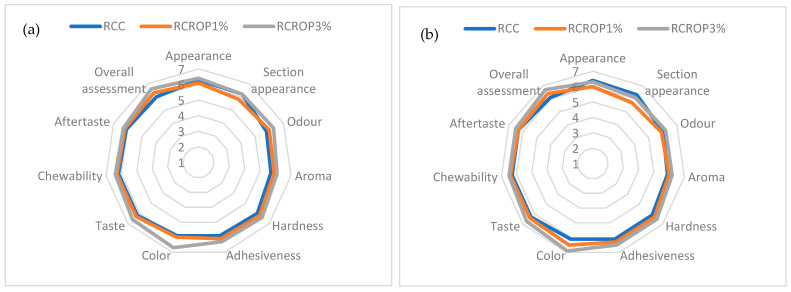
Comparative diagram of the sensory attributes specific to fortified cheeses during storage for 21 days: (**a**) after 1 day, (**b**) after 21 days: RCC cheese without powder addition; RCROP1% and RCROP3% cheese with 1 and 3% powder of ROP.

**Figure 4 foods-13-00182-f004:**
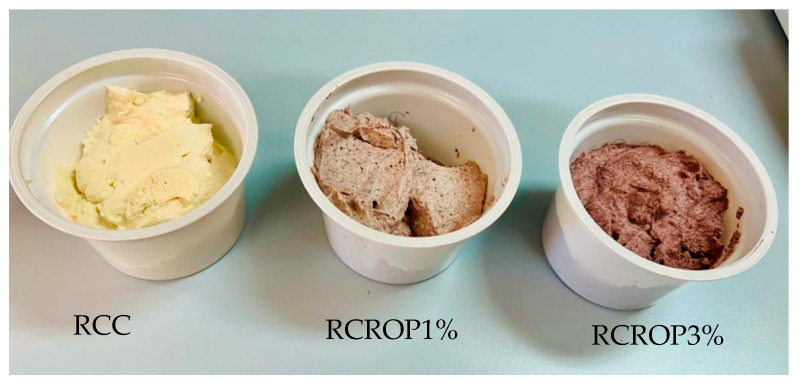
Cheese samples with different percentages of ROP powder: RCC (control), 1% (RCROP1%), and 3% (RCROP3%).

**Table 1 foods-13-00182-t001:** Phytochemical and color properties of ROP powder.

Parameter	ROP Powder
TA, mg C3G/g dw	2.47 ± 0.09
TF, mg QE/g dw	164.66 ± 2.93
TP, mg GAE/g dw	119.69 ± 2.71
Antioxidant activity, µmol TE/g dw	42.29 ± 1.86
Inhibition, %	82.35 ± 1.05
L*	33.08 ± 0.31
a*	21.93 ± 0.22
b*	4.51 ± 0.05
Chroma	22.39 ± 0.22
Hue angle	0.21 ± 0.01
ΔE	39.94 ± 0.27

**Table 2 foods-13-00182-t002:** Phytochemical characteristics and antioxidant activity and stability during 21 days of storage for the control and added-value cheeses.

Phytochemical Characteristics	Storage Time, (Days)	RCC	RCROP1%	RCROP3%
TA, mg C3G/100g dw	0	-	3.43 ± 0.44 ^aA^	6.06 ± 0.52 ^aB^
	21	-	1.92 ± 0.40 ^bA^	3.52 ± 0.48 ^bB^
TP mg GAE/g dw	0	2.56 ± 0.40 ^aA^	4.25 ± 0.46 ^aB^	6.05 ± 0.48 ^aC^
	21	1.61 ± 0.51 ^aA^	3.06 ± 0.59 ^aB^	5.38 ± 0.61 ^aC^
TF mg QE/g dw	0	3.06 ± 0.54 ^aA^	5.13 ± 0.64 ^aB^	7.15 ± 0.55 ^aC^
	21	2.01 ± 0.53 ^aA^	4.06 ± 0.62 ^aB^	6.09 ± 0.54 ^aC^
Antioxidant activity, µmol TE/g dw	0	8.87 ± 0.68 ^aA^	17.08 ± 0.78 ^aB^	24.55 ± 0.67 ^aC^
	21	5.44 ± 0.65 ^bA^	14.15 ± 0.67 ^bB^	21.36 ± 0.65 ^bC^

Values in the table with different superscript lowercase letters show a significant difference (*p* < 0.05) between each tested phytochemical and storage time. A significant difference between each phytochemical parameter and sample variant is indicated by values in the table that have different superscript uppercase letters (*p* < 0.05).

**Table 3 foods-13-00182-t003:** Colorimetric parameters of whey cheese enriched with ROP powder during cold storage for 21 days.

Samples	Storage Time (Day)	L*	a*	b*	Chroma	Hue Angle	ΔE
RCC	0	93.99 ± 0.71 ^aB^	−1.71 ± 0.05 ^aC^	17.64 ± 0.31 ^aD^	17.73 ± 0.29 ^aC^	178.53 ± 0.01 ^aE^	95.65 ± 0.71 ^aA^
	21	90.21 ± 0.42 ^bB^	−1.47 ± 0.41 ^bC^	15.98 ± 0.12 ^aD^	16.05 ± 0.11 ^bC^	178.51 ± 0.02 ^aE^	91.63 ± 0.29 ^aA^
RCROP1%	0	66.23 ± 0.57 ^aA^	9.27 ± 0.11 ^aC^	6.50 ± 0.20 ^aD^	11.32 ± 0.25 ^aB^	0.61 ± 0.01 ^aE^	67.19 ± 0.56 ^aA^
	21	64.52 ± 0.79 ^bA^	12.71 ± 0.32 ^bB^	4.31 ± 0.22 ^bC^	13.49 ± 0.12 ^bB^	0.34 ± 0.02 ^bD^	65.90 ± 0.52 ^bA^
RCROP3%	0	50.72 ± 0.78 ^aB^	13.06 ± 0.19 ^aC^	4.42 ± 0.01 ^aD^	13.79 ± 0.18 ^aC^	0.33 ± 0.01 ^aE^	52.56 ± 0.39 ^aA^
	21	48.67 ± 0.51 ^bB^	17.91 ± 0.68 ^bC^	2.89 ± 0.29 ^bD^	18.14 ± 0.19 ^bC^	0.16 ± 0.04 ^bE^	51.93 ± 0.59 ^bA^

Color parameter variation over time is highlighted by small letters. The color differences between the samples are denoted by capitalized letters. Values that share a lower/uppercase letter are not significantly different (*p* > 0.05).

**Table 4 foods-13-00182-t004:** Textural parameters of the control and fortified cheeses during storage at 5 °C for 21 days: RCC cheese without the addition of ROP powder and RCROP1% and RCROP3% cheese with the addition of 1 and 3% ROP powder.

Parameter	Storage Period (Day)	RCC	RCROP1%	RCROP3%
**Firmness, N**	0	4.34 ± 0.21 ^bA^	5.62 ± 0.30 ^aA^	5.60 ± 0.46 ^aA^
	21	2.92 ± 0.12 ^aA^	3.09 ± 0.22 ^aB^	5.15 ± 0.35 ^bA^
**Adhesion, mJ**	0	0.338 ± 0.011 ^bB^	0.607 ± 0.025 ^aA^	0.596 ± 0.008 ^aA^
	21	0.694 ± 0.023 ^aA^	0.746 ± 0.030 ^aA^	0.608 ± 0.016 ^aA^
**Cohesiveness, -**	0	0.400 ± 0.009 ^aA^	0.340 ± 0.013 ^aA^	0.332 ± 0.019 ^aA^
	21	0.483 ± 0.013 ^aA^	0.471 ± 0.022 ^aA^	0.352 ± 0.008 ^aA^
**Springiness, -**	0	0.445 ± 0.005 ^abA^	0.398 ± 0.015 ^abB^	0.358 ± 0.008 ^aB^
	21	0.592 ± 0.017 ^abA^	0.579 ± 0.021 ^abA^	0.470 ± 0.011 ^aA^
**Gumminess, N**	0	1.738 ± 0.134 ^abA^	1.915 ± 0.177 ^abA^	1.859 ± 0.175 ^aA^
	21	1.391 ± 0.120 ^abB^	1.467 ± 0.164 ^abB^	1.724 ± 0.184 ^aB^
**Chewiness, N**	0	0.782 ± 0.041 ^aB^	0.763 ± 0.073 ^abB^	0.666 ± 0.060 ^abB^
	21	0.830 ± 0.053 ^aA^	0.836 ± 0.081 ^aA^	0.889 ± 0.073 ^aA^
**Resilience, -**	0	0.274 ±0.007 ^bA^	0.219 ± 0.010 ^aA^	0.216 ± 0.010 ^aA^
	21	0.238 ± 0.004 ^bA^	0.210 ± 0.016 ^aA^	0.215 ± 0.007 ^aA^

For each textural parameter and sample, values that do share the same superscript lowercase letter are not significantly different concerning time at *p* > 0.05. Samples that for each textural parameter and storage time do share the same superscript uppercase letter are not significantly different at *p* > 0.05.

**Table 5 foods-13-00182-t005:** Microbial quality of the analyzed cheese samples.

Parameter	RCC	RCROP1%	RCROP3%
TAB	7.53 ± 0.59 ^a^	7.94 ± 0.13 ^a^	8.13 ± 0.26 ^a^
Yeast	0.91 ± 0.12 ^a^	1.48 ± 0.43 ^a^	1.53 ± 0.31 ^a^
Molds	0.43 ± 0.10 ^a^	1.90 ± 0.25 ^a^	2.18 ± 0.29 ^a^
Coliforms	2.47 ± 0.23 ^a^	2.60 ± 0.18 ^a^	2.65 ± 0.25 ^a^
*Escherichia coli*	Not detected	Not detected	Not detected
Coagulase-positive staphylococci	Not detected	Not detected	Not detected

TAB—total aerobic bacteria count. Differences between the analyzed samples were highlighted by lowercase letters per row. Means that share a letter are not significantly different (*p* > 0.05).

## Data Availability

Data is contained within the article.

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
