# Peer review of "Red Onion Peel Powder as a Functional Ingredient for Manufacturing Ricotta Cheese"

_foods, 2024, doi:10.3390/foods13020182_

Round 1

Reviewer 1 Report

Comments and Suggestions for Authors

The submitted manuscript is dedicated to investigating the potential of red onion peel powder as a functional ingredient in whey cheese formulations, as well as the impact of this by-product on the texture, sensory and phytochemical characteristics of fortified cheese. This paper can be useful for food manufacturers who aim to produce healthy food by including natural antioxidants and also offers environmental benefits due to the possibility of reusing agricultural waste. However, I suggest that authors pay attention to my suggestions and comments:

In the abstract: the authors should add more statistical and numerical description of results.

Introduction: Too long introduction, for example Line 40-46, unnecessary for the topic of the paper, shorten it

Line 66-68, List the foods to which onion peel powder has been added so far.

Line 180, …” mg quercetin equivalents (EQ)/g “..per gram of what, did you mean dry substance (dw), write it.

Line 198, Why and how these two additive quantities (1 and 3%) were selected?

Line 343, WCROP1 and WCROP2, Did you mean RCROP1 and RCROP2?

Line 342-343, Emphasize that this also applies to other examined parameters, not only to DPPH.

Table 3, How come a 1% supplement of ROP powder produced darker cheese samples than a 3% supplement?

Line 385-399, Where are the comments for other textural properties like adhesion, cohesiveness, springiness, chewiness?

Line 402-412, Also, discuss the number of aerobic bacteria, yeasts and moulds in the tested samples.

In the results section there are already some parts of the discussion, and then part of the discussion section is expanded, so it is difficult to follow. My recommendation is to merge these two sections.

In the conclusion: The best sample in terms of texture, sensory and phytochemical properties must be expressed.

Author Response

 The authors would like to thank the reviewer for the close reading, and for the proper suggestions and comments aimed to improve the paper. The present version of the paper has been revised, and the specific points have been addressed according to the reviewer's suggestions.

Reviewer 1

Comments

Answers

The submitted manuscript is dedicated to investigating the potential of red onion peel powder as a functional ingredient in whey cheese formulations, as well as the impact of this by-product on the texture, sensory and phytochemical characteristics of fortified cheese. This paper can be useful for food manufacturers who aim to produce healthy food by including natural antioxidants and also offers environmental benefits due to the possibility of reusing agricultural waste. However, I suggest that authors pay attention to my suggestions and comments:

Thank you for your observation. The manuscript has been revised.

In the abstract: the authors should add more statistical and numerical description of results.

Thank you for your observation. The missing information was added to the manuscript as suggested.

Introduction: Too long introduction, for example Line 40-46, unnecessary for the topic of the paper, shorten it.

Thank you for your observation. The sections mentioned were revised in the manuscript as suggested.

Line 66-68, List the foods to which onion peel powder has been added so far.

Thank you for your observation. The information was added in the manuscript.

Line 180, …” mg quercetin equivalents (EQ)/g “..per gram of what, did you mean dry substance (dw), write it.

Thank you for your observation. The missing information was added as suggested.

Line 198, Why and how these two additive quantities (1 and 3%) were selected?

Thank you for your observation.

Proper consideration of the incorporation rate is essential to ensure consistent physical attributes and prevent potential textural defects in the final cheese product.

Prior to this study, preliminary studies were made (data not shown), in which different amounts of red onion peel powder were varied in order to obtain the best quality in terms of appearance, texture, taste, and color.

In general, by increasing the percentage of ROP, the texture, taste, and appearance of enriched cheese samples may be affected leading to differences in cheese texture, consistency, and physical attributes.

The final choice of the respective 1 and 3% concentrations of ROP powder used for the fortification of the studied dairy product had taken into account the positive sensory evaluation of the panelists. Also, sensory attributes allow manufacturers to tailor cheese formulations to meet consumer preferences and deliver a delightful cheese experience.

The marketability of the product was taken into consideration in the stage of the cheese design so a benchmark product named D had the desirable texture that was similar to cheese with a 3% ROP addition.

The recommended addition rate of ROP in the cheese was 1-3% being generally appropriate for enhancing cheese's textural quality.

Line 343, WCROP1 and WCROP2, Did you mean RCROP1 and RCROP2?

Thank you for your observation. The correction was made in the manuscript.

Line 342-343, Emphasize that this also applies to other examined parameters, not only to DPPH.

Thank you for your observation. The information was emphasized as suggested.

Table 3, How come a 1% supplement of ROP powder produced darker cheese samples than a 3% supplement?

Thank you for your observation. Considering all the aspects mentioned, the authors checked and overlooked the database and identified some errors (L*, a* and b* values) in our excel.

Therefore the authors have made all the corrections necessary in the manuscript to express the correct data for the parameters mentioned also, the statistical data was redone.

Line 385-399, Where are the comments for other textural properties like adhesion, cohesiveness, springiness, chewiness?

Thank you for your observation. The information was added as suggested.

Line 402-412, Also, discuss the number of aerobic bacteria, yeasts and moulds in the tested samples.

Thank you for your observation. The information was added as suggested.

In the results section there are already some parts of the discussion, and then part of the discussion section is expanded, so it is difficult to follow. My recommendation is to merge these two sections.

Thank you for your observation. The two sections (section 3 results and section 4 discussions) were merged as suggested.

In the conclusion: The best sample in terms of texture, sensory and phytochemical properties must be expressed.

Thank you for your observation. The missing information was added at lines 570-572.

Reviewer 2 Report

Comments and Suggestions for Authors

This study determines the effect of red onion skin powder addition on the functional properties and potential pro-health activity of ricotta cheese. The utilization of edible agricuture by-products, especially those rich in bioactive compounds follows current trends in food technology, thus, the topic of this study seems quite interesting to potential readers. The abstract well reflects the content of the manuscript, but more specific results could be added. The introduction provides a good background of the study and includes relevant references. The experiments are well designed and described in sufficient details. Complex evaluation of obtained product features deserves particular attention. The modes of results presentation are clear; however, some parts of the results section are more suitable for the discussion section (i.e. lines 301-315, 343-351, 422-424, 447-448). Authors should order it. Dhe discussion is supported by the observations and obtained results of the study. The conclusion summarizes the most important findings. Despite the some described technical nature gaps, which sholud be corrected, the quality of the manuscript is quite satissfactory. Therefore, I recommended minor revision.

Detailed suggestions:

Line 196, 212, 219, 228, 237 – the term „value-added” seems to be very speculative during the methodology description. Maybe „supplemented” or „fortified”?

Line 210 – Why different abbreviations were used for the 1% and 3% supplements?  i.e. ROP vs. ROPS Please, check it (i.e. ROP vs. ROPS); Line 319 – Please explain the abbreviation, when it is first time used. I also recommended unifying abbreviations, for example as in line 210 (i.e. RCROP1% and RCROP3%).

Line 422 – add refernce.

Author Response

The authors would like to thank the reviewers for their close reading and for the proper suggestions and comments aimed at improving the paper. The present version of the paper has been revised, and the specific points have been addressed according to the reviewer's suggestions.

 Reviewer 2

This study determines the effect of red onion skin powder addition on the functional properties and potential pro-health activity of ricotta cheese. The utilization of edible agriculture by-products, especially those rich in bioactive compounds, follows current trends in food technology; thus, the topic of this study seems quite interesting to potential readers. The abstract well reflects the content of the manuscript, but more specific results could be added. The introduction provides a good background on the study and includes relevant references. The experiments are well designed and described in sufficient detail. The complex evaluation of the obtained product features deserves particular attention. The modes of results presentation are clear; however, some parts of the results section are more suitable for the discussion section (i.e., lines 301-315, 343-351, 422-424, and 447-448). Authors should order it. The discussion is supported by the observations and results of the study. The conclusion summarizes the most important findings. Despite the some described technical nature gaps, which should be corrected, the quality of the manuscript is quite satisfactory. Therefore, I recommended a minor revision.

Thank you for your observation. The manuscript has been revised.

Detailed suggestions:

Lines 196, 212, 219, 228, and 237—the term „value-added” seems to be very speculative during the methodology description. Maybe „supplemented” or „fortified”?

Thank you for your observation. The correction was made through the manuscript as suggested.

Line 210 – Why different abbreviations were used for the 1% and 3% supplements?  i.e. ROP vs. ROPS Please, check it (i.e. ROP vs. ROPS); Line 319 – Please explain the abbreviation, when it is first time used. I also recommended unifying abbreviations, for example as in line 210 (i.e. RCROP1% and RCROP3%).

Thank you for your observation. The correction was made through the manuscript as suggested.

Line 422 – add reference.

Thank you for your observation. The reference was added as suggested.
